# INTEREST-BASED ITEM REPRESENTATION FRAMEWORK FOR RECOMMENDATION WITH MULTI-INTERESTS CAPSULE NETWORK

## ABSTRACT

Item representation plays an important role for recommendation, such as e-commerce, news, video, etc. It has been used by retrieval and ranking model to capture user-item relationship based on user behaviors. For recommendation systems, user interaction behaviors imply single or multi interests of the user, not only items themselves in the sequences. Existing representation learning methods mainly focus on optimizing item-based mechanism between user interaction sequences and candidate item(especially attention mechanism, sequential modeling). However, item representations learned by these methods lack modeling mechanism to reflect user interests. That is, the methods may be less effective and indirect to capture user interests. We propose a framework to learn interest-based item representations directly by introducing user Multi Interests Capsule Network(MICN). To make the framework model-agnostic, User Multi Interests Capsule Network is designed as an auxiliary task to jointly learn item-based item representations and interest-based item representations. Hence, the generic framework can be easily used to improve existing recommendation models without model redesign. The proposed approach is evaluated on multiple types of benchmarks. Furthermore, we investigate several situations on various deep neural networks, different length of behavior sequences and joint learning ratio of interest-based item representations. Experiment shows a great enhancement on performance of various recommendation models and has also validated our approach. We expect the framework could be widely used for recommendation systems.

## 1 INTRODUCTION

With the rapid development of deep learning, great achievements have been made in recommendation, such as news recommendation, video recommendation, e-commence and advertisement. For recommendation systems, user interaction behaviors imply single or multi interests of the user, not only items themselves in the sequences. In general, users may have multi interests, e.g., a user interacts with products from several different categories, including clothes, sports and food. The interests lay below the interactive behaviors which increases the difficult of capturing them directly.

For recommendation systems, how to target different users' interests is key object. Series of deep learning models on click-through rate (CTR) prediction have been proposed. Wide & Deep (Cheng et al., 2016) jointly trains wide linear models and deep neural networks to combine the benefits of memorization and generalization for recommender systems. PNN, Deep Crossing (Shan et al., 2016) and DeepFM (Guo et al., 2017) try to extract low-order and high-order feature extraction by adopting a product layer. DIN(Zhou et al., 2018) uses attention mechanism to increase the pooling weights of similar items. DIEN (Zhou et al., 2019) introduces sequential model to build the sequential character instead of using item embedding directly in DIN. DIEN extracts hidden states of GRU as attention input and uses AUGRU to take place of traditional attention model. Generally, deep neural networks depict user interests from previous user-item interactions by utilizing item embedding vectors.

To settle the diffusion matter of interests, (Sabour et al., 2017) proposes dynamic routing capsule network and (Edraki et al., 2020) successfully achieves the better understanding of relationship between objects than CNNs. Based on dynamic routing of capsule network, (Li et al., 2019) proposes

MIND for dealing with user's diverse interests in retrieval systems by representing one user with multiple vectors encoding the different aspects of the user's interests.

However, existing representation learning methods mainly focus on optimizing item-based mechanism between user interactive behavior sequences and candidate item, therefore ignore the impacting of label diffusing into each item in user interactive behavior sequences which weaken the intensity of user interests in back propagation of training model. Besides, with the network layers getting deeper and the dimension of embedding layer becoming larger, new method always need to redesign the model architecture, bring in new dataset or other information. Hence, a method is necessary to enhance the model performance and could be generally used in practice without redesigning whole model architecture or bringing extra information.

In this paper, we propose a framework to learn interest-based item representations directly by introducing user Multi Interests Capsule Network(MICN). To make the framework model-agnostic, Multi Interests Capsule Network is designed as an auxiliary task((Pi et al., 2019) introduces a auxiliary task to enhance the model performance) to jointly learn item-based item representations and interest-based item representations. Interest-based item representation generated by MICN shared with original model takes user diverse interest information in whole model.

The contributions of this paper can be summarized as follows:

• A new framework to learn interest-based item representations by introducing user Multi Interests Capsule Network(MICN). MICN is designed as auxiliary task and easily integrated with original recommendation model.

• The new item representation is generated by concatenating interest-based item representation produced by MICN and item-based item representation produced by original model.

• An approach of joint learning method and hyper parameter optimization with MICN framework. Experimental results show the great improvement of different recommendation models on benchmark datasets.

We do experiments on different public datasets and compare results between original ranking model (such as Wide & Deep, DIN, DIEN) and models with auxiliary MICN. The results demonstrate the framework we proposed has better performance than the original model.

## 2 MODEL ARCHITECTURE

In recommendation system, $i$ denotes the item id in practice, $\mathbb{I}_u$ denotes a set of user interactive behavior sequences (clicked or viewed item sequences) by an user u. $p_u$ is basic user profile information. $i_t$ is the candidate item id and $r_t$ is related information of candidate item from recommendation system. $f(i_t, r_t)$ is the feature function of candidate item information from candidate item $i_t$ and item related information $r_t$.

Usually, user interests is represented to learn the function $f(\mathbb{I}_u, p_u)$, including user profile $p_u$ and user interactive behavior sequences $\mathbb{I}_u$. Hence the user interests can be formulated as

$$V_u = f(\mathbb{I}_u, p_u) \tag{1}$$

where $V = (v_u^1, v_u^2, \ldots, v_u^K) \in \mathbb{R}^{h \times K}$ is the representation vector learning from user u information $\mathbb{I}_u$ and $p_u$, $h$ is user interest vector dimension and $K$ is number of user vector dimension. Actually, $v_u^1$ is represent one of user multi interests vector and $V$ is the collection of them.

Additionally, candidate item representation is to learn the function $f(i_t, r_t)$ of item id and it's information. We can obtain the candidate item embeddings by

$$e_t = f(i_u, r_t) \tag{2}$$

where $e_t \in \mathbb{R}^h$ is the representation embedding vector learn from target item id $i_t$ and it's related information $r_t$. In fact, $e_t$ always is the Embeddings & Pooling layer vector taken from User Multi Interest Capsule Networks which present in next section.

Recommendation system is push 'good' item for a visiting web or app user. Hence, the score of measuring the relationship between candidate item and user interests is necessary. Then the score is define as

$$f_{score}(V_u, e_t) = e_t^T v_u^k \tag{3}$$

We can obtain a value $f_{score}$ to measure the distance between user interests and candidate items. Finally, according to the collection of user multi interests, recommendation system will select the top 'good' items for user.

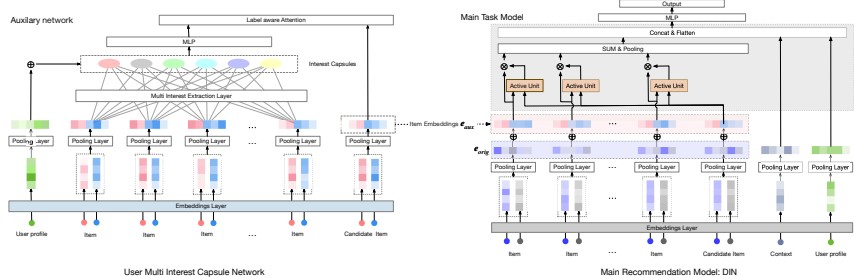

Figure 1: Model architecture of user Multi Interest Capsule Network(MICN) as auxiliary task and share item embeddings with main model task(DIN). MICN takes user interactive behavior sequences and user profile into dynamic routing capsule and extracts user interests as capsules. Scaled dot product between interest capsule and candidate item embeddings makes them in the same representation space. In label aware attention layer. The item embedding vector combined with item embeddings of original recommending model brings user diverse interest-based item representation in main model. The main model is designed suitable for specific task, such as ranking model etc. DIN model structure showed as an example and item embeddings $e_{orig}$ has extended with interest-based item embeddings $e_{aux}$. Then the input embeddings of main model is $e$ in Equation 10. The MICN helps main task promoting the performance without change the model original architecture or bring in other dataset or information.

## 2.1 USER MULTI INTERESTS CAPSULE NETWORKS

to circumvent some limitations of CNNs, capsules replace scalars with vectors to encode appearance feature representation, allowing better preservation of spatial relationships between whole objects and its parts. They also introduced the dynamic routing mechanism, which allows to weight the contributions of parts to a whole object differently at each inference step.

The multi interests of user usually hide in interactive behavior sequences and profile information. Capsules(Sabour et al., 2017) replace scalars with vectors to encode appearance feature representation by assemble a group of neurons. Dynamic routing of capsule network learn the weight of different capsule which capable of encoding the relationship between the part and the whole. Capsule has better understanding of relationship between objects than CNNs (Edraki et al., 2020). In recommendation system, MIND (Li et al., 2019) automatically capture the high-level multi interests of user through dynamic routing of capsule and achieve good performance in retrieval system of e-commence. Consequently, we propose item representation based on user multi interest capsule networks(MICN) to help CTR prediction model promoting their performance.

We briefly introduce dynamic (Behavior to Interest) routing of capsule for learning the representation of multi interests from user profile information and interactive behavior sequences. For the input of each capsule

$$v_j = \frac{\|s_j\|^2}{1 + \|s_j\|^2} \frac{s_j}{\|s_j\|} \tag{4}$$

where $v_j$ is output and $s_j$ is all of input of capsule $j$.

$$s_j = \sum_i c_{ij}\hat{x}_{j|i} = \sum c_{ij}W_{ij}x_j \tag{5}$$

$$c_{ij} = \frac{\exp(b_{ij})}{\sum_k \exp(b_{ik})} \tag{6}$$

Then through the dynamic routing to capture the high-level abstract interests from raw features of user. $c_{ij}$ is softmax function for input $b_{ij}$. The behavior to interest(B2I) (Li et al., 2019) adaptively

aggregate user's view sequences into multi interests representing vectors. According to the routing logits Equation 3, the $b_{ij}$ is defined as

$$b_{ij} = \boldsymbol{u}_j^T \boldsymbol{S} \boldsymbol{e}_i, \qquad i \in \boldsymbol{I}, j \in \{1, 2, \ldots, K\} \tag{7}$$

where $\boldsymbol{e}_i \in \mathbb{R}^h$, and $ve_i$ is one of item $i$ embedding vector of user interactive behavior sequences. $\boldsymbol{u}_j \in \mathbb{R}^h$, $j \in \{1, 2, \ldots, K\}$ the capsule vector of user interest, $K$ is hyper parameter which is the number of user's interests. $\boldsymbol{S} \in \mathbb{R}^{h \times h}$ the bilinear mapping matrix, link the user's capsule interests and viewed sequences. $b_{ij}$ is connection on user's interest and item and keep them in the same vector space mapping.

Though capture multi interest capsule vectors from user interactive behavior sequences and profile information, MIND introduced label-aware attention based on scaled dot product to measure the relationship between user's interest and item information. In label aware attention layer, candidate item is query and user interest capsule is key and value, candidate item embedding vector is represented in interest capsule space. Then the scaled dot product formulate as

$$\boldsymbol{v}_u = \boldsymbol{V}_u \text{softmax}(pow(\boldsymbol{V}_u^T \boldsymbol{e}_i, p)) \tag{8}$$

Consequently, we obtain the probability $P(\boldsymbol{e}_i | \boldsymbol{v}_u)$ and use softmax activate function to select 'Good' one. The training loss is defined as

$$L_{micn} = \sum_{u,i} \log P(\boldsymbol{e}_i | \boldsymbol{v}_u) \tag{9}$$

where $L_{micn}$ is the loss of user multi interests capsule network loss. As match between user's interests and candidate item, the item embeddings vector and user's interest capsule vector have the same vector space which based on the user's interests representation, which is very important for interest-based item representation.

## 2.2 Interest based Item Embeddings Representation

Embedding representation based on deep learning is of much concern in practice(Wang et al., 2020). In recommendation system, each recommending model has its own generating embeddings method. Many works introduced in Section 1 extract multi interests represented by item embeddings through designing network structures. Further, the impact of label diffusing into each item in user interactive behavior sequences weaken the intensity of user interests in back propagation of training model. Though dynamic routing capsule network can partially settle the diffusion matter of interests, specifically integrated with recommending model needing to redesign the model architecture is difficult and not generally used. Inspired by DIEN, auxiliary task play significant role in improving model performance. In order to refrain from redesigning the complex main task model architecture, an auxiliary task is introduced for better item representation learning. Therefore, we propose a model framework of user Multi Interest Capsule Network(MICN) as auxiliary task and share interest based item embeddings with main recommendation model task which is Deep Interest Network(DIN). According to the Equation 7, the scaled dot product, the distance between interest and item, makes item embedding vector is indicated by user interest capsule vectors. Besides, auxiliary task brings item embeddings expressed by user interest capsule in main model by sharing the item embedding vector. In main model, we define the item embeddings compose of two parts:

$$\boldsymbol{e} = \boldsymbol{e}_{orig} \oplus \boldsymbol{e}_{aux} \tag{10}$$

where $\oplus$ is concatenation operator, $\boldsymbol{e}_{orig}$ and $\boldsymbol{e}_{aux}$ are item embedding vector correspondingly in the main target recommending model task and auxiliary task designed as MICN. This framework not only expands original model item embeddings referring user interest capsule, but also keep the original model architecture and still have the original model property. Consequently, the framework can be applied in general recommending model.

Because of the item embedding vector of main recommending model is combination of original model item embeddings and auxiliary model item embeddings, original item embeddings is not influenced by auxiliary task and auxiliary item embedding is influence by two task which can be controlled by a hyper parameter. Hence, the total loss of whole model is formulate as

$$L = L_{main} + \lambda L_{micn} \tag{11}$$

Table 1: Experimental Datasets

| Datasets | Users | Items | Categories | Samples |
|---|---|---|---|---|
| Books | 603,668 | 367,982 | 1,600 | 603,668 |
| Electronics | 192,403 | 63,001 | 801 | 192,403 |

Table 2: Experimental Results(AUC) on Industrial Datasets

| Model | Electronics | Books |
|---|---|---|
| Wide&Deep | $0.7461 \pm 0.0015$ | $0.7860 \pm 0.0013$ |
| **Wide&Deep-MICN** | **0.7502 $\pm$ 0.0010** | **0.7928 $\pm$ 0.0009** |
| DIN | $0.7569 \pm 0.0009$ | $0.7970 \pm 0.0010$ |
| **DIN-MICN** | **0.7606 $\pm$ 0.0013** | **0.8002 $\pm$ 0.0009** |
| DIEN | $0.7706 \pm 0.0021$ | $0.8534 \pm 0.0018$ |
| **DIEN-MICN** | **0.7723 $\pm$ 0.0002** | **0.8633 $\pm$ 0.0019** |

where $L$ is the total loss of the whole model, $L_{main}$ is the loss of main model and $L_{micn}$ is the loss of user interest capsule network. $\lambda$ is the hyper parameters which adjust the balance of loss and auxiliary task loss.

For the $L_{micn}$, label aware attention layer need positive samples in constructing the loss function. Hence negative samples of label item are masked so that make the loss work when training the whole model. During model training process, the item embedding $e$ will receive two parts of back propagating gradient: main task and auxiliary task. $e_{orig}$ only receive main model gradient $\nabla e_{main}$ and $e_{aux}$ receive auxiliary model(MICN) gradient $\nabla e_{main}$ as well as main original model gradient $\nabla e_{aux\_main}$ in order to make $e_{aux}$ fit for main model. The gradient of auxiliary task is update following $\nabla e_{aux} = (1 - \varphi)\nabla e_{aux\_aux} + \varphi\nabla e_{aux\_main}$ where $\varphi \in [0, 1]$. How to choose the suitable hyper parameter $\varphi$ will be discussed in the experiment.

## 3 EXPERIMENTS

In this section, Experiments using user multi interest capsule network(MICN) we proposed as auxiliary task conducted on Amazon real world dataset and the results will be showed and discussed.

### 3.1 DATASETS

We use real world dataset from Amazon Electronics and Books datasets (Ni et al., 2019) to verify our proposed network architecture. Table 1 showed the basic information including user, item and category information.

**Amazon Datasets**

Amazon Customer Reviews (a.k.a. Product Reviews) is one of Amazon's iconic datasets. Millions of Amazon customers have committed hundreds million reviews of their opinions and shopping experiences regarding products on the Amazon.com website.

**Electronics**: Including 192403 consumer reviews for Amazon electronics products like HDMI cables, bluetooth speakers etc, 63001 products covering 801 categories and total 1,689,188 samples. The dataset includes basic product information, category for each product and viewed products list.

**Books**: List of 603668 consumer reviews for Amazon books, 367,982 books covering 1600 categories and total 603668 samples. The dataset includes basic product information, category and viewed products list.

### 3.2 EXPERIMENT AND RESULTS

We present experiment of our model framework on the above Amazon dataset Electronics and Books comparing with other widely used model in industrial field.

Table 3: Hyper parameter $\varphi$ in DIEN with multi interest capsule network as auxiliary task on Amazon books dataset.

| $\varphi$ | AUC | $\varphi$ | AUC |
|---|---|---|---|
| 0.1 | $0.8620 \pm 0.0018$ | 0.6 | $0.8606 \pm 0.0014$ |
| 0.2 | $0.8636 \pm 0.0019$ | 0.7 | $0.8625 \pm 0.0010$ |
| 0.3 | $0.8638 \pm 0.0005$ | 0.8 | $0.8624 \pm 0.0018$ |
| 0.4 | $0.8636 \pm 0.0013$ | 0.9 | $0.8621 \pm 0.0012$ |
| 0.5 | $0.8606 \pm 0.0012$ | 1.0 | $0.8602 \pm 0.0022$ |

Table 4: The AUC performance of MICN framework with DIEN ranking model on various length of user interactive behavior sequences on Amazon books dataset.

| length | 10 | 20 | 50 |
|---|---|---|---|
| DIEN | $0.8543 \pm 0.0016$ | $0.8534 \pm 0.0018$ | $0.8505 \pm 0.0022$ |
| DIEN-MICN | $0.8571 \pm 0.0018$ | $0.8633 \pm 0.0019$ | $0.8613 \pm 0.0014$ |

Wide & Deep(Cheng et al., 2016): combines DNN as Deep part and linear model as Wide part to achieve model generalization and memorization.

DIN(Zhou et al., 2018): uses attentional mechanism modeling relationship between user interactive behavior sequences and candidate item and obtain contribution of each item made in historical visiting sequences to the target item.

DIEN(Zhou et al., 2019): based on DIN model architectures, adds GRUs to modeling the evolutional interests of users with attention.

We implement those model described above on Amazon opened industrial dataset. Then we add an auxiliary task introduced in Section 2 to accomplish multi interests extraction of user and compare the improvement of performance. We set one layer of multi interests extraction layer contains $K = [\log_2 |\boldsymbol{I}_u|]$ dynamic interest capsules, make the interest-based item embedding vector from MICN shared with main recommendation model and gradient updated by a small ratio $\varphi = 0.3$. As total loss combines main task loss and auxiliary loss, the weight hyper parameter of loss $\lambda$ is 1. Optimizer is Adam and learning rate is 1e-4.

Each experiment is repeated 5 times. The mean value and standard deviation of model performance AUC are calculated to measure the model performance. Table 2 showed the performance of different model architecture. In Amazon datasets, maximal length of consumer viewed electronic products list is 10 and maximal length of consumer viewed books is 20. Obviously, the performance of all the model architecture is enhanced by the framework which indeed demonstrates the advantages of MICN as an auxiliary task, meanwhile, the main recommendation model still keep original structure.

Furthermore, with the increase length of viewed items, the framework has better effect on longer interactive behavior sequences. Thus, we conduct the experiment comparing the performance between the framework we proposed and benchmark on different length of user viewed sequences. Table 4 showed the experimental result. DIEN with MICN have better AUC result than original DIEN model. Consequently, for long click sequences of user, user interests capsule network works more efficient and accuracy to extract diverse user interests which conducted different type of experiments to verify.

## 4 CONCLUSION AND FUTURE WORKS

In this paper, we focus on a generic framework of interest-based item embedding representation to enhance performance of recommendation model. A framework to learn interest-based item representations directly by introducing user multi-interests capsule network was proposed. User multi interest capsule network is designed as an auxiliary task to jointly learn item-based item representations and interest-based item representations. The framework can be easily generalized to recommendation model architecture without extra data or substantial change of model. Besides, the

framework has an effect on longer viewed sequences and more diverse interests. With experimental result on industrial datasets, the framework of user multi interest capsule network as auxiliary task outperforms in the terms of click-throughout rate prediction task. In the future, we will try to extend our work to explainable recommendation of users' interests through multi interest capsule network.

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
