# OpenReview forum: "Interest-based Item Representation Framework for Recommendation with Multi-Interests Capsule Network"
_ICLR.cc/2022/Conference — ICLR 2022 Submitted_

### Official Review · Reviewer_LCWN · 2021-10-29

**Correctness:** 1
**Technical Novelty And Significance:** 2
**Empirical Novelty And Significance:** 2
**Recommendation:** 3
**Confidence:** 5

**Main Review:**

Strong points

S2. This paper is easy to understand.

S3. The model architecture is simple and easy to understand.

Weak points

W1. The baseline models are not state-of-the-art models. I recommend that the authors consider recent publications for the CTR problem as below.
-	[A] Yuan Cheng, Yanbo Xue, Looking at CTR Prediction Again: Is Attention All You Need? SIGIR 2021
-	[B] Weiyu Cheng, Yanyan Shen, Linpeng Huang, Adaptive Factorization Network: Learning Adaptive-Order Feature Interactions, AAAI 2020
-	[C] Ruoxi Wang, Rakesh Shivanna, Derek Z. Cheng, Sagar Jain, Dong Lin, Lichan Hong, Ed H. Chi, DCN V2: Improved Deep & Cross Network and Practical Lessons for Web-scale Learning to Rank Systems, WWW 2021

W2. It should be more explanation for experimental results. The authors should analyze the meaning of experimental results.

W3. The technical novelty is incremental and unclear.

Minor issue.
-	Figure 1 is too small to recognize.


**Summary Of The Paper:**

This paper proposes a user multi-interests capsule network (MICN) to improve better item representation. Because the proposed model is model-agnostic, it can be combined with existing models. The intuition of the proposed model is unclear. The proposed model does not consider the state-of-the-art CTR models. In this sense, I recommend that the authors clearly argue the key novelty of the proposed model and extensive experimental setup.

**Summary Of The Review:**

In terms of technical novelty and experimental setup, the quality of this paper is not enough to be accepted to ICLR.

---

### Official Review · Reviewer_KquS · 2021-10-30

**Correctness:** 2
**Technical Novelty And Significance:** 2
**Empirical Novelty And Significance:** 2
**Recommendation:** 1
**Confidence:** 4

**Main Review:**

Strength:
The authors conduct experiments on multiple datasets.

Weaknesses:
1. The motivation of this work is problematic or unconvincing. It is not clear why the auxiliary interest-based item representations are necessary because they can be inherently encoded by item-based user interest models. The authors do not provide evidence on this point. In addition, it is not clear why the training task is also the recommendation task. It seems that the proposed method is in fact a kind of ensemble.
2. The contribution of this paper is also marginal. The authors directly adopt the off-the-shelf capsule network MIND for multi-interest modeling. There are also many other existing multi-interest modeling works [1-3]. The authors do not cite nor compare any other multi-interest modeling methods.
3. The evaluation is rather flawed. The authors only compare the results of adding MICN to several methods. However, there are many methods that aim to enhance item embeddings in existing literature (like [1]). In addition, the authors do not evaluate the effectiveness of the MICN method. Thus, it is not clear whether the improvement is brought by the multi-interest architecture or simply brought by the increase of model complexity.
4. Many important details are missing. For example, many hyperparameters in this work are not included, and the methods for constructing training and evaluation samples are not introduced.
5. The coverage of related work is too limited. The authors should track the recent advances in the recommender system community.
6. The writing and presentation also need improvement. The motivation part needs revisions most. Many sentences are not clear. The experimental discussions are uninformative. There are many typos and grammatical errors. The fonts in figure 1 are too small.


[1] Pal, Aditya, et al. "PinnerSage: multi-modal user embedding framework for recommendations at pinterest." Proceedings of the 26th ACM SIGKDD International Conference on Knowledge Discovery & Data Mining. 2020.
[2] Qi, Tao, et al. "HieRec: Hierarchical User Interest Modeling for Personalized News Recommendation." arXiv preprint arXiv:2106.04408 (2021).
[3] Liu, Zheng, et al. "Octopus: Comprehensive and Elastic User Representation for the Generation of Recommendation Candidates." Proceedings of the 43rd International ACM SIGIR Conference on Research and Development in Information Retrieval. 2020.

**Summary Of The Paper:**

This paper introduces an interest-based item representation learning method with a multi-interest capsule network. The authors use an auxiliary task to learn interest-based item representations, which are further combined with item representations learned from their features for final recommendation. Experiments on two domains of the Amazon product dataset and an industrial dataset show that the proposed method outperforms several baseline methods.

**Summary Of The Review:**

The motivation of this work is not clear. This paper lacks sufficient technical contribution and the evaluation is also not thorough. The coverage of related work is quite limited. The writing also needs improvement. Thus, my recommendation is a strong rejection.

---

### Official Review · Reviewer_wjxV · 2021-11-01

**Correctness:** 3
**Technical Novelty And Significance:** 1
**Empirical Novelty And Significance:** 2
**Recommendation:** 3
**Confidence:** 4

**Main Review:**

Strengths:
- Important task: This paper focuses on dealing with CTR prediction, which is a very practical task that highly relates to profit in the industry.
- Well-organized: This paper contains basic elements that are essential to a research paper.

Weaknesses：
- contribution: The main framework MICN proposed in this paper serves as an extension to existing frameworks. However, MICN is mostly borrowed from a previous work called MIND (Li et al., 2019), and MIND can not be regarded as the main contribution of this submission.
- Motivation is not clear: It is not clear the reason this paper uses capsule networks. Why are feedforward networks not able to serve the same purpose?
- Weak baselines: The baselines included in the experiment part are not very strong. Since CTR prediction is a well-studied problem, I suggest the authors review more strong related works, such as [1] et al.
- Case studies missing: It would be interesting for the reviewer to see more details in the experiments, such as sensitivity w.r.t. Gamma, some visualization of the learned correlation between user interests and candidate items,
Writing can be improved: I find it hard to understand some sentences when reviewing the draft. I suggest the authors proofread the draft more and fix all grammar issues and typos.

References:
- [1] Fei, Hongliang, et al. "GemNN: Gating-enhanced Multi-task Neural Networks with Feature Interaction Learning for CTR Prediction." Proceedings of the 44th International ACM SIGIR Conference on Research and Development in Information Retrieval. 2021.


**Summary Of The Paper:**

This paper studies the item recommendation task. It claims that existing studies ignore the correlation between user interests and candidate items and that they are not easily extendable. To deal with this,  an auxiliary task fulfilled by Multi Interests Capsule Network (MICN) is proposed to extend the existing frameworks. Experiments are conducted via the click-through rate (CTR) prediction. The MICN can improve very slightly on the Amazon books dataset.

**Summary Of The Review:**

This paper studies a practical yet well-studied problem by applying existing techniques. The reviewer finds the paper well-organized. However, this paper has limited contribution, insufficient experiments, and missing details. These major drawbacks prevent the reviewer from championing accepting the paper.  In summary, I suggest rejecting the paper at ICLR and that the authors make major revisions until their next submission.

---

### Official Review · Reviewer_P2Vg · 2021-11-03

**Correctness:** 3
**Technical Novelty And Significance:** 2
**Empirical Novelty And Significance:** 2
**Recommendation:** 3
**Confidence:** 3

**Main Review:**

Strength:
1. Novel model to jointly learn of item-based and interest-based representation of the item for recommended systems
2. Practical application in recommended systems with real datasets.

Weaknesses:
1. Combination of the state-of-the-art models i.e., Capsules(Sabour et al., 2017) and MIND (Li et al., 2019), etc.
2. There is no theoretical justification to support the advancement of the proposed model compared to the previous techniques.
3. There should be more empirical results to support the claims. It would be more interesting to see the real recommendation rather than the AUC (yet another :)).

**Summary Of The Paper:**

The paper proposes an interest-based item embedding representation to enhance the performance of recommendation model by jointly learning iitem-based item representations and interest-based item representations.

**Summary Of The Review:**

As Above

---

### Decision · Program_Chairs · 2022-01-20

**Decision:**

Reject

**Comment:**

This paper proposed a joint learning approach which combines item-based representations and interest-based representations to improve recommender systems. Overall the scores are negative among all the reviewers. The reviewers acknowledge that the proposed approach provides a simple yet effective way to improve the existing item-based representations. However, all the reviewers pointed out concerns around the motivation and limited novelty (the proposed approach mostly combines a few existing approaches together without careful examination/exploration in the experiments). Furthermore, the baselines considered in the paper are on the relatively weak side. The authors didn't provide any response. Therefore, I vote for reject.